# Carfilzomib-Based Regimen and Cardiotoxicity in Multiple Myeloma: Incidence of Cardiovascular Events and Organ Damage in Carfilzomib-Dexamethasone versus Carfilzomib-Lenalidomide-Dexamethasone. A Real-Life Prospective Study

**DOI:** 10.3390/cancers15030955

**Published:** 2023-02-02

**Authors:** Anna Astarita, Giulia Mingrone, Lorenzo Airale, Marco Cesareo, Anna Colomba, Cinzia Catarinella, Dario Leone, Francesca Gay, Sara Bringhen, Franco Veglio, Alberto Milan, Fabrizio Vallelonga

**Affiliations:** 1Hypertension Unit, Department of Medical Sciences, Division of Internal Medicine, AO “Città Della Salute e Della Scienza” University Hospital, 10126 Turin, Italy; 2Myeloma Unit, Department of Medical Sciences, Division of Hematology, AO “Città Della Salute e Della Scienza” University Hospital, 10126 Turin, Italy

**Keywords:** carfilzomib, cardiotoxicity, Kd, KRd, real-life, dexamethasone, lenalidomide, multiple myeloma, cardiovascular adverse events, hypertensive events, echocardiography, pulse wave velocity

## Abstract

**Simple Summary:**

Cardiotoxicity is a known adverse effect of Carfilzomib therapy; nevertheless, limited data are available on the comparison of the cardiovascular complications induced by Carfilzomib-dexamethasone versus Carfilzomib-lenalidomide-dexamethasone in patients with multiple myeloma (MM) in a real-life setting. We conducted a prospective study to determine differences in incidence and time of onset of hypertensive- and major cardiovascular-adverse events between in-patients with MM treated with the two regimens. Furthermore, we investigated differences in subclinical cardiac and vascular organ damage in these two groups, which might benefit from different monitoring strategies.

**Abstract:**

Carfilzomib-mediated cardiotoxicity in multiple myeloma (MM) is a well-established adverse effect, however limited data are available on the comparison of cardiovascular complications in patients treated with Carfilzomib-dexamethasone (target dose of K 56 mg/m^2^) versus Carfilzomib-lenalidomide-dexamethasone (target dose of K 27 mg/m^2^) beyond controlled trials. A total of 109 patients were enrolled, 47 (43%) received Kd and 62 (57%) KRd. They then underwent a baseline and follow-up evaluation including trans-thoracic echocardiography and arterial stiffness estimation. All types of cardiovascular and hypertensive events occurred more frequently in the Kd group compared with the KRd (59% vs. 40% and 55% vs. 35.5% patients, respectively, *p* ≤ 0.05), with higher incidence of hypertensive. The time of onset of any type of CVAE, and of major and hypertensive events was shorter in the Kd regimen (*p* ≤ 0.05). At follow-up, Kd patients more frequently developed signs of cardiac (decline of global longitudinal strain) and vascular organ damage (rise of pulse wave velocity), as compared with KRd. Despite the older age, longer history of MM and longer period of pre-treatment of Kd patients, these factors did not increase the probability of incidence for all types of cardiovascular events at multivariate analysis (*p* > 0.05). In conclusion, the Kd regimen showed greater cardiovascular toxicity and earlier onset of events with respect to KRd. Thus, a closer and thorough follow-up should be considered.

## 1. Introduction

Carfilzomib (K) is a next-generation proteasome inhibitor approved for relapsed/refractory multiple myeloma (RRMM) and as an induction/consolidation therapy of new MM diagnoses [1]. The introduction of K-based chemotherapy regimens has dramatically improved the prognosis and the survival rate of MM patients [2,3]. However, its use is associated with significant cardiotoxic effect, manifesting either during the course of therapy or after completion of treatment [4,5,6,7,8].

Among the K-based regimen, Carfilzomib-dexamethasone (Kd) and Carfilzomib-lenalidomide-dexamethasone (KRd) have been approved for the treatment of RRMM, based on interim results of the phase III ENDEAVOR and ASPIRE studies [1,9]. In the ENDEAVOR study, Kd regimen (K administered at the dose of 20 mg/m^2^ on days one and two, and at 56 mg/m^2^ thereafter) was superior to the active comparator—bortezomib-dexamethasone—in improving median progression-free survival (PFS) and median overall free survival (OS) in RRMM patients [1]. Similarly, the ASPIRE study demonstrated the superiority of KRd (K administered at the dose of 20 mg/m^2^ on days one and two, and at 27 mg/m^2^ thereafter) versus lenalidomide-dexamethasone in PFS and disease response [9].

However, the use of K-based protocols in MM patients presents many challenges, first of all the risk of cardiotoxicity, including heart failure, ischemic heart disease, arterial hypertension and death due to cardiovascular events [1,6,9].

The data on Kd and KRd therapy and cardiac toxicity are derived from the abovementioned phase III trials, on a highly selected population of patients under optimal management conditions. Little is known regarding the differences in incidence of such toxicities between the two regimens in the real world, due to the heterogeneity of the criteria used to classify the nature of cardiovascular toxicity. Although more events are expected in patients receiving higher doses of carfilzomib, few data derived from real-life settings are available. Furthermore, no different protocols for follow-up have been assumed based on the administered dose of K. 

The aim of this perspective study was to investigate the difference in incidence and time of onset of cardiovascular adverse events between Kd and KRd in MM patients in real-life conditions. Furthermore, we investigated differences in subclinical organ damage (both cardiac and vascular) induced by the two regimens and offer a discussion on how these data might support different follow-up strategies for patients treated with Kd or KRd regimen. 

## 2. Materials and Methods

This prospective study was conducted at the Echocardiography Laboratory of the the Hypertension Unit in collaboration with the Myeloma Unit Division of ‘Città della Salute e della Scienza’ Hospital in Turin, Italy. Adults with MM, who had haematological indication to Kd or KRd, were consecutively enrolled. Patients with a prior K-based treatment or light chain cardiac amyloidosis (assessed by end-organ biopsy or cardiac magnetic resonance) were excluded.

### 2.1. Carfilzomib-Based Regimen

Patients in the Kd group received K as a 30-min infusion at a starting dose of 20 mg/m^2^ on day 1 and 2, and then as a target dose of 56 mg/m^2^ on day 8, 9, 15, and 16 of the first cycle; on subsequenst cycles the drug was administered on day 1, 2, 8, 9, 15 and 16. All patients additionally received dexamethasone 20 mg on day 1, 2, 8, 9, 15, 16, 22, and 23 of each 28-day cycle. 

Similarly, patients in the KRd group received K as a 10-min infusion at a starting dose of 20 mg/m^2^ on day 1 and 2, and then as a target dose of 27 mg/m^2^ on days 8, 9, 15, and 16 of the first cycle; on subsequenst cycles the drug was administered on day 1, 2, 8, 9, 15, and 16. Lenalidomide was given orally at the standard dose of 25 mg on day 1 through 21. Dexamethasone was given at the dose of 40 mg on day 1, 8, 15, and 22.

Dose reductions were decided by the Hematology specialist on a case by case basis for patients who had adverse events.

### 2.2. Assessments

Details of the study methodology have been reported previously [4]. Briefly, in accordance with the European Myeloma Network Protocol [10], all patients who had indication for Kd or KRd underwent a baseline cardiovascular assessment before treatment initiation. This included a clinical visit with anamnestic focus of cardiovascular past medical history and risk factors, laboratory testing (including cardiac serum biomarkers), office blood pressure measurements, ambulatory blood pressure monitoring (24 h-ABPM), 12-leads EKG, trans-thoracic echocardiography with global longitudinal strain assessment (GLS), and estimation of arterial stiffness by the carotid–femoral pulse wave velocity (cfPWV) measurement. During the first visit, patients with office and/or out-of-office blood pressure values in the high/normal range and those with known arterial hypertension were advised to start anti-hypertensive treatment or to optimize a previous anti-hypertensive therapy, in order to obtain an optimal blood pressure control before starting K therapy [11]. K-based regimen, timing and dosing were decided by the hematology specialists.

The above described clinical, laboratory and instrumental evaluation (with the exclusion of 24 h-ABPM) was repeated after 6 months of K-based treatement and/or at the time of any suspected CVAEs. Data on CVAEs were collected during the follow-up, by integrating the clinical data (if second evaluation was performed) with the periodic review of patients’ haematologic reports and phone interviews.

### 2.3. CVAEs

All types of cardiovascular adverse events were considered, and grouped into major cardiovascular events and arterial hypertensive events (see Appendix A for definition of CVAEs). The following were considered as major cardiovascular events: acute coronary syndromes, heart failure, arrhythmias, typical chest pain, syncope, sudden death, new onset left ventricle dysfunction—defined as reduction in left ventricle ejection fraction—and/or relative decline of GLS value from baseline, according to the current guidelines on management and prevention of cardiotoxicity [12]. 

Hypertensive events included: new diagnosis of arterial hypertension or the worsening of known arterial hypertension, uncontrolled hypertension prior to K-infusion, uncontrolled hypertension following K-infusion, masked hypertension, hypertensive urgency, hypertensive emergency [11].

CVAEs were graded according to the National Cancer Institute Common Terminology Criteria for Adverse Events, version 5.0 [13].

### 2.4. Statistical Analysis

Differences in baseline parameters between Kd and KRd patients were investigated by the chi-square test/Fisher’s exact test for categorical variables and by the unpaired t-test/Mann–Whitney test for continuous variables, as appropriate. A two-sided *p* value < 0.05 was used as the level of statistical significance. Hazard ratios were estimated with a stratified Cox proportional hazards model and distributions were summarized with the use of the Kaplan–Meier method. The analyses were performed using a dedicated software (IBM SPSS Statistics, Version 22.0.0.0, IBM Corp., Armonk, NY, USA).

## 3. Results

### 3.1. Baseline Characteristics and Cardiovascular Risk Profile

Between March 2015 and July 2022, a total of 109 MM patients were enrolled: 47 patients (43%) designated as the Kd group, 62 patients (57%) as the KRd group. Figure 1 shows the study protocol.

Both Kd and KRd groups showed a high cardiovascular risk profile and a significant prevalence of subclinical cardiac and vascular organ damage at baseline, exemplified by left ventricle hypertrophy (19% and 16%; *p* = 0.681), left atrial enlargement (34% and 24.2%; *p* = 0.259), GLS impairment (25% and 15.3%; *p* = 0.216) and pulse wave velocity ≥ 9 m/s (29.5% and 28.3%; *p* = 0.893).

Notably, up to 55% of Kd patients and 46% of KRd patients had elevated blood pressure values (based on office and 24-ABPM), with a large proportion of patients not reporting a prior history of hypertension (newly diagnosed HTN in one forth to almost one third of patients in the two groups). In about 40% of patients in each group, the initiation of a new anti-hypertensive treatment or the uptitration of previous therapy was needed. Baseline characteristics were balanced between treatment groups, with the exception of LV EF, which was slightly lower in Kd patients (Table 1).

### 3.2. CVAEs: Incidence and Time of Onset of All-Type-, Major- and Hypertensive-Events

All patients received at least one dose of the study chemotherapy and were followed-up for a mean time of 14 months.

All types of cardiovascular and hypertensive events occurred more frequently in Kd patients (Table 2). A total of 28 patients receiving Kd (55%) experienced at least one cardiovascular event as compared with 25 patients (40%) receiving KRd (*p* = 0.046). Adverse events of grade 3 or higher were reported in 36.2% patients receiving Kd and in 22.6% patients receiving KRd. In a multivariate analysis the incidence of all types of cardiovascular events between groups did not change after the correction for the age of patients at baseline, the duration of MM disease or the number of prior pre-treatments (HR 1.02, *p* = 0.384; HR 1.00, *p* = 0.972; HR 0.95, *p* = 0.664, respectively).

Similarly, 26 (55.3%) patients receiving Kd and 22 (35.5%) receiving KRd experienced at least one hypertensive event (*p* = 0.039), with higher incidence of hypertensive urgencies in Kd group (*p* = 0.004).

Major cardiovascular events were not significantly different overall among the two groups (25% vs. 14% for patients receiving Kd vs. KRd, *p* = 0.149). Incidence of major cardiovascular events in Kd versus KRd patients were: acute coronary syndromes (4.2 vs. 1.6%), typical chest pain (6.3 vs. 1.6%), heart failure (10.6 vs. 3.2%), syncope (0 vs. 1.6%), arrhythmias (8.5 vs. 8.0%), sudden death (0 vs. 1.6%), LVEF impairment (4.2 vs. 0%, two patients in the Kd group, one of which had a new LVEF reduction <40% and one of which had a new LVEF reduction by ≥10 percentage points to an LVEF of 40–49%), and a new relative decline in GLS (8.5% vs. 1.6%). Table 2 reports the incidence of each CVAEs in the study population.

Furthermore, Kd treatment was associated with an increased risk of all-type, major and hypertensive event occurrence compared with KRd, as shown in Figure 2.

Analysing the time of onset of the first CVAE, we observed a shorter time of onset of all-type, major and hypertensive events in the Kd group. Specifically comparing Kd with KRd, the first all-type cardiovascular event was observed at a mean time of 3.6 vs. 10.2 months (*p* < 0.001), the first hypertensive event at 4.0 vs. 10.8 months (*p* < 0.001) and the first major cardiovascular event at 8.6 vs. 14.6 months (*p* = 0.005). 

The median duration of the treatment was 6.5 months for patients receiving Kd and 10.6 months for patients receiving KRd. A total of 91.4% of patients receiving Kd and 80.6% of patients receiving KRd discontinued treatment. The discontinuation was most commonly due to disease progression (67.4 and 32% respectively for Kd and KRd). Seven (14.9%) patients receiving Kd and two (3.2%) receiving KRd interrupted the therapy due to cardiotoxicity.

### 3.3. Cardiac and Vascular Organ Damage Induced by the Therapy: Kd vs. KRd 

A total of 88 patients, 34 receiving Kd and 54 receiving KRd, underwent follow-up clinical assessment at a median time of 5.7 months and 5.1 months respectively from treatment initiation (Table 3). Office BP values were similar in the two groups; however, on average more Kd patients were advised to increase their anti-hypertensive treatment between the first and the second visit, due to intercurrent hypertensive events (mean number of HTN medications: 2.2 vs. 1 in Kd vs. KRd recipient, *p* = 0.013). Nevertheless, based on blood pressure values at second visit, up to 55% of patients receiving Kd had the indication to further increase their anti-hypertensive treatment.

Subclinical cardiac organ damage as detailed above was investigated by trans-thoracic echocardiography, including GLS assessment. Patients receiving Kd had worse GLS values compared with patients receiving KRd (−19.7 ± 2.9 vs. −21.3 ± 2.4; *p* = 0.009). LV EF was lower at baseline for the group of patients receiving Kd treatment and such a difference was maintained with a further reduction in LV EF in patients treated with Kd (58.9 ± 5.9 vs. 63.0 ± 5.7; *p* = 0.002). 

Conversely, patients receiving KRd showed higher prevalence of left ventricle hypertrophy (11 vs. 30%; *p* = 0.046), while no significant differences in left ventricle mass was observed. No other differences were observed for the remaining echocardiographic parameters. 

The median cumulative dose of K at the time of the second echocardiography was 2775.5 mg/m^2^ and 1332.7 mg/m^2^ respectively in patients receiving Kd and KRd (*p* < 0.001). For patients receiving Kd, the cumulative dose at the time of the second echocardiography was statistically correlated with GLS value (r = 0.44; *p* = 0.026) and was associated with a relative decline of GLS ≥ 3 unit from the baseline on Cox regression analysis (OR = 0.9; *p* = 0.029). Finally, for subclinical vascular organ damage, patients receiving Kd had a statistically significant rise of PWV compared with patients receiving KRd (8.5 ± 1.9 vs. 7.4 ± 1.5; *p* = 0.009). Figure 3 summarizes the main findings regarding subclinical organ dagame at follow up for the two treatment populations.

## 4. Discussion

Despite the considerable efficacy in patients with multiple myeloma of Kd and KRd regimens, K has been associated with a high risk of cardiovascular complications, which in turn contributes to disease-associated morbidity and mortality [4,6,7,14,15]. Cardiovascular adverse events in MM patients treated with K based regimens are the result of an overlapping of individual risk factors (age, pre-existent cardiovascular diseases, etc.), myeloma risk factors (renal failure, amyloidosis, hyper viscosity, etc.) and the direct effect of K, through induction of cardiomyocytes apoptosis and disruption of nitric oxide homeostasis in endothelial cells [16,17]. Data on the direct comparison of cardiotoxicity induced by K within Kd and KRd protocols in multiple myeloma patients, prospectively followed, in a real-life setting, are limited. Although a higher incidence of cardiovascular events is expected in Kd patients—due to the higher target dose of K 56 mg/m^2^ in Kd versus 27 mg/m^2^ in KRd protocol and to higher grade of frailty—few studies have compared the cardiotoxicity induced by the two treatments in a real-life setting. Hence the appropriate clinical management of these patients often represents an area of uncertainty. 

The present study is a prospective, real-life investigation focused on the adverse cardiovascular effects of Kd and KRd regimens. The study followed the protocol of the European Hematology Association and the European Myeloma Network [10] for the cardiovascular risk assessment in this specific population, as previously prospectively validated [4]. Following the protocol, we confirmed the high cardiovascular risk profile of MM patients undergoing treatment with Kd and KRd in a real-life setting, as demonstrated by the high prevalence of cardiovascular risk factors, pre-existent cardiovascular diseases, and subclinical cardiac and vascular organ damage. Notably, in our study the latter was similar between groups at baseline, despite the longer multiple myeloma duration in patients receiving Kd. Notably, the combination of office blood pressure and 24-h ABPM, in our experience made possible the diagnosis of HTN in 26% of the study population, hence allowing for the initiation of treatment of an otherwise unrecognized and highly impactful CV risk factor in these patients. Given the high cardiovascular risk of patients scheduled for Kd and KRd therapy in real-life settings and the recognized cardiotoxicity of this medication, the correct estimation of the cardiovascular risk before starting the treatment is central [4]. 

A number of cardiovascular adverse events was observed in all patients, with a higher incidence in patients receiving Kd, although the median duration of the treatment was shorter in patients receiving this regimen. All-type cardiovascular events and hypertensive events occurred more frequently in patients receiving Kd versus those receiving KRd. However, despite the high rate of CVAEs observed, only a small number of patients required discontinuation of K-based therapy or a reduction of drug dosage. Regarding the hypertensive adverse events, the severity of cases was similar between groups, except for the hypertensive urgencies, which were more frequent in patients receiving Kd. However, it should be noted that these patients were advised to increase their anti-hypertensive treatment between the first and the second evaluation, due to intercurrent hypertensive events. Nevertheless, based on blood pressure values at second visit, the same group of patients had the indication to further uptitrate the anti-hypertensive treatment. In addition, patients taking part in the KRd protocol received higher doses of dexamethasone, which is a known vasopressor. These findings suggest a greater negative effect on blood pressure values of the Kd protocol. As such, a more aggressive anti-hypertensive therapy should be considered for patient planning of Kd treatment after their baseline evaluation, to reduce the risk of incident subclinical organ damage and cardiovascular events (myocardial infarction, stroke, etc.). Although the rate of major CV events was not significantly different in the two groups of patients, it was numerically higher in patients receiving Kd—the low number of cases might explain the lack of statistical significance and this finding should be considered hypothesis generating. Despite the main baseline parameters being similar between Kd and KRd patients, the greater incidence of cardiovascular complications observed might be affected by confounding factors including the older age, longer history of MM and longer period of pre-treatment in patients of the Kd group with respect to those in the KRd group. In this regard, in a multivariate analysis, the presence of these factors did not increase the probability of incidence of all-type cardiovascular events observed in the two groups. However, patient candidates for the Kd protocol represent a more fragile category of patients that require a closer follow-up, particularly if treated with higher doses of K. 

The incidence of all-type cardiovascular events observed in the present study differs with respect to Onda et al. [18], a retrospective comparative study between Kd and KRd in MM patients, in which the frequency of CVAEs reported was lower in both groups (9.3 vs. 4% of cases of heart failure, 0.9 vs. 4% cases of arterial hypertension, 0.9 vs. 0% of arrhythmias). These differences might be explained by the real-life setting of the present study, the specific focus on cardiovascular events, and the prospectively echocardiography evaluation of cardiac function, which are different to most oncology studies. Furthermore, the comparison of the incidence of CVAEs between studies is limited by the heterogeneity of the criteria used to classify the nature of cardiovascular events.

A particular point of interest was the observation of the shorter time of onset of all-type, major and hypertensive events in patients receiving Kd compared with patients on the KRd regimen. Notably, the Kd protocol is based on a higher K dose, thus suggesting a dose dependency of the K cardiac and vascular toxicity. The follow-up clinical and instrumental assessment allowed the early recognition of incidental cardiac and vascular organ damage. According to current guidelines on cardiotoxicity prevention and management [12], the cardiac damage was evaluated through trans-thoracic echocardiography. A relative decline in mean GLS values was evident in patients receiving Kd. Similarly, LV EF—notably lower at baseline for patients in the Kd group—was further reduced in these patients at follow up. These data are consistent with the results of the ENDEAVOUR cardiac substudy [1], that reported an incidence of 3.7% for LV EF impairment in patients receiving Kd during the course of treatment. No data are available in the literature on the impact of KRd treatment on LV EF. Similarly, no studies are available on the impact of either regimens on GLS or other echocardiographic parameters of subclinical organ damage, although the role of GLS as a prognostic factor for adverse events and as a marker of early left ventricle systolic function impairment induced by K therapy is well recognized [4,7,19,20]. PWV, indirect marker of arterial stiffness and an independent predictor of adverse cardiovascular events and organ damage [4,5], did rise more during treatment in patients treated with Kd, probably due to the higher doses of the drug. 

At present, no studies have addressed the optimal follow-up scheme in patients treated with K [12]. Our results suggest the need for a careful baseline assessment and of different follow-up plans based on the type of K-based protocol, with a closer surveillance likely appropriate for patients scheduled for Kd treatment and, in general, for regimens based on higher doses of K (>56 mg/m^2^). Furthermore, the grade of frailty at baseline and during the follow-up must be carefully evaluated, especially for patient candidates for the Kd regimen, who are usually older and more pre-treated. 

Our study has several limitations. Firstly, the small sample of the cohort and the experience of a single institution did not allow us to generalize the results, which must therefore be confirmed by larger studies with a multicentre design. The low number of cases might explain the lack of statistical significance of incidence of major cardiovascular events and of other subclinical organ damage, these findings should be considered hypothesis generating. For the same reason, the incidence of hypertensive events observed may have been affected both by the optimization of anti-hypertensive treatment prior to chemotherapy initiation and by the accurate detection of all hypertensive events during treatment, due to the specific area of expertise of our single centre (Hypertension Unit), which might make our results less generalizable to a different clinical setting. Finally, the absence of a control arm disallowed the detection of potential confounding factors. Another limitation was the absence of standardized dosing of the K-based regimen (due to reduction of the dose or missed infusions because of adverse events/other reasons). 

## 5. Conclusions

Among K-based chemotherapy regimens for MM, the Kd regimen showed a greater cardiovascular toxicity compared with the KRd regimen, based on a higher incidence of cardiovascular complications and subclinical cardiac and vascular organ damage during the course of treatment. Furthermore, cardiovascular adverse events occurred earlier in patients treated with Kd. Our findings reinforce and extend the evidence of the central role of a careful cardiovascular assessment in MM patients prior to, and during, K-based treatment, and moreover for a closer surveillance in patients scheduled for Kd treatment and, in general, for regimens based on higher doses of K.

## Figures and Tables

**Figure 1 cancers-15-00955-f001:**
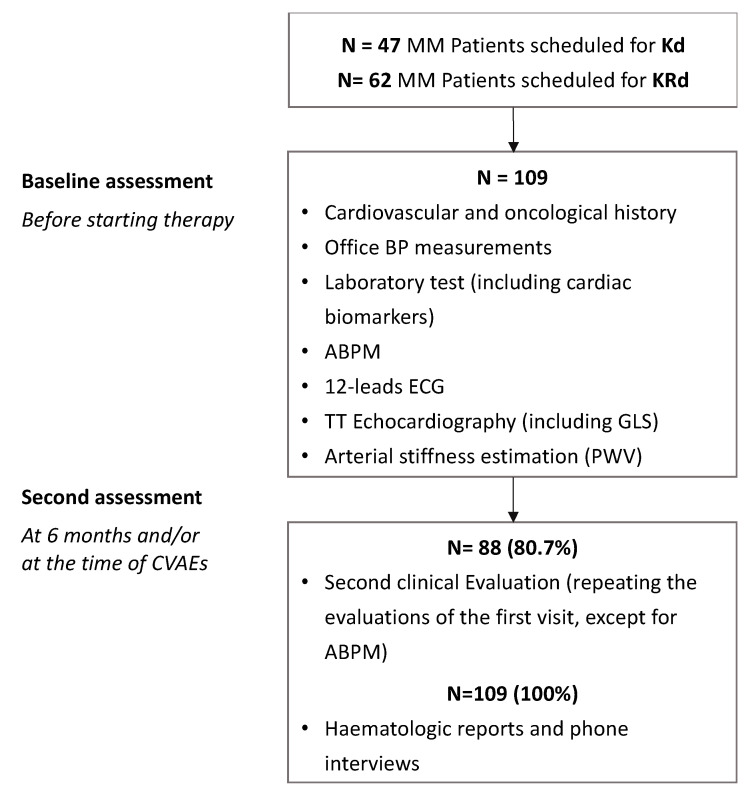
Study protocol.

**Figure 2 cancers-15-00955-f002:**
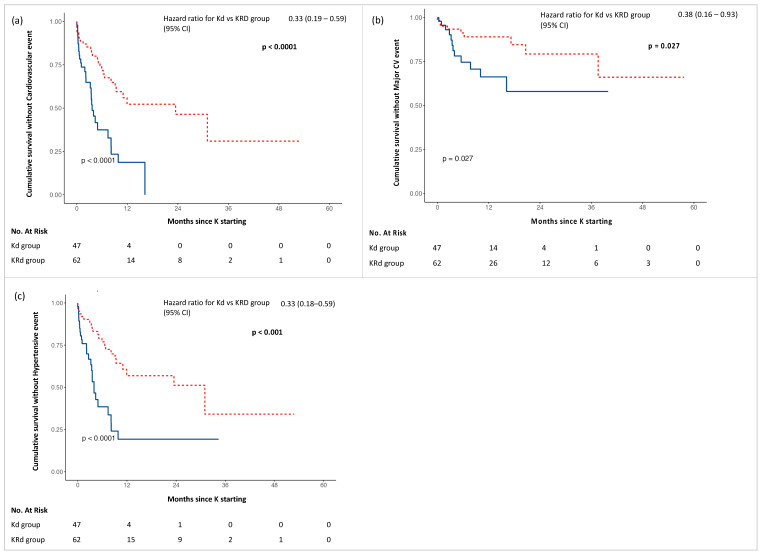
Kaplan–Meier curves for adverse events in Kd (blue) and KRd (red) groups: (**a**) Cumulative survival without Cardiovascular event; (**b**) Cumulative survival without Major CV event; (**c**) Cumulative survival without Hypertensive event.

**Figure 3 cancers-15-00955-f003:**
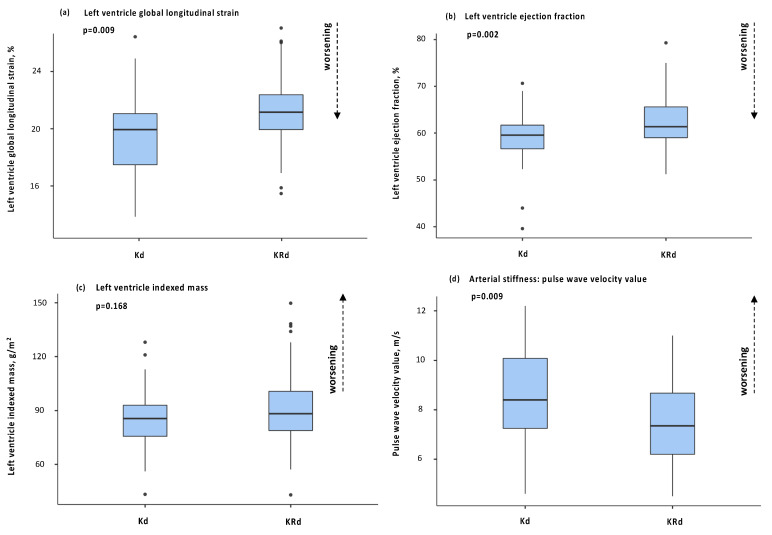
Box plots of main parameters of organ damage at second visit between Kd and KRd: (**a**) left ventricle global longitudinal strain; (**b**) left ventricle ejection fraction; (**c**) left ventricle indexed mass; (**d**) pulse wave velocity.

**Table 1 cancers-15-00955-t001:** Baseline population characteristics.

Characteristic	KD No. 47 (43%)	KRD No. 62 (57%)	*p*
**General**			
Age, mean (SD), years	68.1 ± 8.5	66.8 ± 8.3	0.431
Male sex, No. (%)	30 (63.8)	29 (46.8)	0.077
**Multiple myeloma**			
MM disease duration, mean (SD), months	89.2 ± 56.4	52.6 ± 46.7	<0.001
ISS stage, No. (%)			
ISS I-II	15 (31.9)	35 (56.4)	0.290
ISS III	7 (14.8)	7 (11.2)	0.279
ISS not reported	25 (53.1)	20 (32.2)	
Prior regimens, mean (SD)	2.9 ± 1.5	1.3 ± 0.6	<0.001
Previous therapies, ˆ No. (%)			
Anthracyclines	9 (21.4)	11 (19)	0.761
Alkylating agents	34 (81)	48 (82.8)	0.817
Immunomodulating agents	41 (97.6)	38 (65.5)	<0.001
Proteasome inhibitors	34 (81)	56 (94.9)	0.026
**CV diseases**, No. (%)			
Tobacco use (prior/current)	25 (53.2)	30 (48.8)	0.619
Obesity (BMI ≥ 30)	10 (21.3)	19 (30.6)	0.273
Known arterial hypertension	23 (48.9)	28 (45.2)	0.696
Diabetes	6 (12.8)	6 (9.7)	0.610
Chronic renal failure (eGFR < 60 mL/m)	15 (33.3)	13 (24.1)	0.308
Ischemic heart disease	2 (4.3)	1 (1.6)	0.404
Atrial fibrillation	3 (6.5)	1 (1.6)	0.211
Dyslipidaemia	5 (10.6)	11 (17.7)	0.299
Previous stroke	1 (2.1)	1 (1.6)	0.843
**Office BP**			
SBP, mean (SD), mmHg	128.8 ± 18.1	129.9 ± 18.3	0.749
DBP, mean (SD), mmHg	76.9 ± 12.6	75.7 ± 11.1	0.616
**ABPM ***			
Daytime SBP, mean (SD), mmHg	125.2 ± 14.1	125.5 ± 12.7	0.930
Daytime DBP, mean (SD), mmHg	74.7 ± 9.9	74.2 ± 9.4	0.780
24 h SBP, mean (SD), mmHg	121.3 ± 13.6	121.1 ± 12.3	0.956
24 h DBP, mean (SD), mmHg	71.7 ± 9.3	70.8 ± 8.5	0.626
24 h MBP, mean (SD), mmHg	88.4 ± 10.5	88.4 ± 9.0	0.966
Night-time SBP, mean (SD), mmHg	111.0 ± 13.9	110.7 ± 13.4	0.785
Blood pressure variability, mean (SD)	9.0 ± 3.0	9.0 ± 3.2	0.818
Diagnosis of arterial hyp. at ABPM, No (%)	21 (46.7)	23 (42.6)	
**Arterial Hypertension Profile**			
Uncontrolled BP values at first assessment (office and/or ABPM), No. (%)	26 (55.3)	28 (46.7)	0.374
High-normal BP values arterial at first assessment, No. (%)	4 (8.5)	5 (8.3)	0.974
New diagnosis of hyp., No.(%)	11 (23.4)	18 (29)	0.565
Indication to increase anti-hypertensive therapy, No (%)	21 (44.7)	26 (41.9)	0.774
Number of anti-hypertensive drugs after first assessment, mean (SD)	1.5 ± 1.8	1.2 ± 1.1	0.293
**Echocardiography**			
Left ventricle morphology			
LVMi, mean (SD), g/m^2^	90.0 ± 24.3	85.5 ± 20.6	0.303
LVH, No (%)	9 (19.1)	10 (16.1)	0.681
RWT, mean, (SD)	0.44 ± 0.1	0.43 ± 0.09	0.809
Left ventricle function			
LVEF, mean (SD), %	60.7 ± 6.8	63.8 ± 6.1	0.010
LVEF reduction, No (%)	3 (6.4)	2 (3.3)	0.447
GLS value ^†^, mean (SD), %	−21.3 ± 2.3	−22.2 ± 2.2	0.063
GLS value ≥ −20%, No (%)	11 (25)	9 (15.3)	0.216
Diastolic dysfunction, No (%)	4 (8.5)	3 (4.8)	0.439
TDI E’ sept, mean (SD), cm/s	6.7 ± 2.0	6.8 ± 1.7	0.444
TDI E’ lat, mean (SD), cm/s	8.4 ± 2.1	9.1 ± 2.5	0.125
E/E’ avg, mean (SD)	8.4 ± 1.9	8.5 ± 2.6	0.881
Left atrium			
LAVi, mean (SD), mL/m^2^	29.5 ± 9.2	28.5 ± 9.3	0.579
LAVi enlargement, No. (%)	16 (34)	15 (24.2)	0.259
Right ventricle			
TAPSE, mean (SD), mm	24.7 ± 4.6	23.9 ± 4.9	0.314
PAPs, mean (SD), mmHg	20.9 ± 9.5	18.2 ± 8.7	0.099
Pulmonary hypertension, No (%)	18 (38.3)	18 (29)	0.308
**Arterial stiffness estimation**			
cfPWV value, mean (SD), m/s	8.0 ± 2.0	8.0 ± 1.7	0.974
cfPWV value ^‡^ ≥ 9 m/s, No (%)	13 (29.5)	15 (28.3)	0.893

Mean values estimated in: * 99 patients; ^†^ 103 patients; ^‡^ 97 patients. ˆ patients were mostly treated with multiple therapies, hence total % might amount to > 100. MM = multiple myeloma; ISS = MM international staging system; BMI = body mass index; eGFR = estimated glomerular filtration rate; SBP = systolic blood pressure; DBP = diastolic blood pressure; HR = heart rate; ABPM = ambulatory blood pressure monitoring; MBP = mean blood pressure; LAVi = left atrial volume indexed to body surface area; LVMi = left ventricular mass indexed to body surface area; LVEF = left ventricle ejection fraction; LVH = left ventricular hypertrophy (male LVMi ≥ 105 g/m^2^; female LVMi ≥ 95 g/m^2^); RWT = relative wall thickness; GLS = global longitudinal strain; cfWV = carotid–femoral pulse wave velocity; hyp. = hypertension.

**Table 2 cancers-15-00955-t002:** Cardiovascular adverse events in Kd vs. KRd group.

	KD	No. CVAEs		KRD	No. CVAEs		*p*
	No. 47	Grade 1–2	Grade 3–4	No. 62	Grade 1–2	Grade 3–4	
**Major CVAEs**							
*No. of patients*	12 (25.5)			9 (14.5)			0.149
*No. of events **							
ACS (STEMI)	1	0	1	0	0	0	
ACS (NSTEMI)	1	0	1	1	0	1	
Typical chest pain	3	0	3	1	0	1	
Heart failure	5	3	2	2	0	2	
Syncope	0	0	0	1	0	1	
Arrhythmias	4	2	2	5	3	2	
Sudden death	0	NA	0	1	NA	1	
LVEF impairment	2	0	2	0	0	0	
GLS impairment	4	NA	NA	1	NA	NA	
**Hypertensive CVAEs**							
*No. of patients*	26 (55.3)			22 (35.5)			0.039
*No. of events **							
New onset/worsened HTN	24	24	0	17	17	0	
Masked HTN	1	1	0	1	1	0	
Pre-infusion HTN	24	13	11	18	8	10	
Post-infusion HTN	7	5	2	8	4	4	
HTN urgency	6	0	5	0	0	0	
HTN emergency	0	0	0	0	0	0	
**All-type CVAEs**							
*No. of patients*	28 (59.6)			25 (40.3)			0.046

* Patients experienced more than one CVAE, hence total % amount > 100. CVAEs = cardiovascular adverse events; ACS = acute coronary syndrome; STEMI = ST-elevation myocardium infarction; NSTEMI = Non-ST elevation myocardial infarction; HTN = hypertension; LVEF = left ventricle ejection fraction; GLS = global longitudinal strain; NA = not applicable.

**Table 3 cancers-15-00955-t003:** Main parameters and subclinical organ damage at the second assessment.

Characteristic	KD No. 34 (38.6%)	KRD No. 54 (61.3%)	*p*
**Office BP**			
SBP, mean (SD), mmHg	129.4 ± 16.5	122.9 ± 15.1	0.184
DBP, mean (SD), mmHg	75.9 ± 11.6	71.7 ± 8.8	0.062
**Arterial Hypertension Profile**			
Office uncontrolled BP values, No. (%)	10 (29.4)	10 (18.5)	0.235
High-normal BP values, No. (%)	5 (14.7)	6 (11.1)	0.620
Indication to increase anti-hypertensive therapy at second assessment, No (%)	19 (55.9)	13 (24.5)	0.003
Number of anti-hypertensive drugs before the second assessment, mean (SD)	2.2 ± 1.6	1.0 ± 1.2	0.013
**Echocardiography**			
Left ventricle morphology			
LVMi, mean (SD), g/m^2^	85.3 ± 17.1	91.4 ± 21.5	0.168
LVH, No (%)	4 (11.8)	16 (30.2)	0.046
RWT, mean, (SD)	0.43 ± 0.1	0.43 ± 0.1	0.832
Left ventricle function			
LVEF, mean (SD), %	58.9 ± 5.9	63.0 ± 5.7	0.002
LVEF reduction, No (%)	2 (5.9)	0 (0)	0.074
GLS value ^†^, mean (SD), %	−19.7 ± 2.9	−21.3 ± 2.4	0.009
GLS value ≥ −20%, No (%)	13 (43.3)	13 (24.5)	0.076
Diastolic dysfunction, No (%)	7 (20)	6 (11.3)	0.261
TDI E’ sept, mean (SD), cm/s	5.5 ± 1.5	6.1 ± 1.7	0.109
TDI E’ lat, mean (SD), cm/s	7.9 ± 2.7	8.4 ± 2.1	0.360
E/E’ avg, mean (SD)	9.5 ± 3.5	9.9 ± 3.2	0.440
Left atrium			
LAVi, mean (SD), mL/m^2^	29.9 ± 12.7	31.6 ± 9.3	0.458
LAVi enlargement, No. (%)	13 (38.2)	20 (37.7)	0.963
Right ventricle			
TAPSE, mean (SD), mm	22.7 ± 4.3	23.4 ± 3.9	0.489
Pulmonary hypertension, No (%)	13 (38.2)	17 (32.1)	0.555
**Arterial stiffness estimation**			
cfPWV value, mean (SD), m/s	8.5 ± 1.9	7.4 ± 1.5	0.009
cfPWV value ^‡^ ≥ 9 m/s, No (%)	13 (43.3)	6 (17.6)	0.025

Mean values estimated in: ^†^ 83 patients; ^‡^ 64 patients. SBP = systolic blood pressure; DBP = diastolic blood pressure; LAVi = left atrial volume indexed to body surface area; LVMi = left ventricular mass indexed to body surface area; LVEF = left ventricle ejection fraction; LVH = left ventricular hypertrophy; GLS = global longitudinal strain; cfPWV = carotid–femoral pulse wave velocity.

## Data Availability

The data can be shared up on request.

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
