# Peer review of "Carfilzomib-Based Regimen and Cardiotoxicity in Multiple Myeloma: Incidence of Cardiovascular Events and Organ Damage in Carfilzomib-Dexamethasone versus Carfilzomib-Lenalidomide-Dexamethasone. A Real-Life Prospective Study"

_cancers, 2023, doi:10.3390/cancers15030955_

Round 1
Reviewer 1 Report
The paper by Astarita et al. with strong hematology support of Drs. Gay and Bringhen, compared the cardiotoxicity of Kd vs. KRd in "real life settings", and albeit being of interest, needs to be substantially improved, because otherwiese extremely misleading in its tone.
The below comments will support the authors in achieving this:
1. In the abstract, the authors need to correct, that there are no "few" but plenty of cardiovascular complication data, including in real-life settings, i.e. even EMN and Italian/German guidelines, that have published recommentations, "nevtheless updated reports are certainly warrented"!
2. Cave should be urged by the authors already in stating that they prospectively (or rather retrospectively assessed) cardiovascular events in Kd vs. KRd patients, because both patient groups were extremely different, i.e. on the Kd-group a) much more pretreted, b) had received prior IMIDs in almost 100% and c) were deliberately treated rather with a 2-agent than 3-agent combo, albeit d) much higher K-dose, thus the likelihood of the Kd-group to indeed aquire and show cardiac events much more frequently than the Krd MM group of patients: very much anticipated.
Since the Kd vs. KRd dose was much different, thus the aimed 56mg/m2 dose vs. 20/27 or 36mg/m2 dose used in the KRd arm much reduced for the latter than in the former group.
If that is not stated throughout the paper and already in the abstract, which should induce a more down-to-earth interpretation of the data, I would refrain from publishing this article, because otherwise less experienced MM colleagues migh indeed "understand" that KRd is less toxic than Kd, which if equal K-doses were used, is certainly not correct.
3. Before any cardiac differenced of both patient groups are described in the abstract, the main differences in patient charactersistics must be discribed (i.e. see 2.)
4. Cave with english phrasing and many typos/spelling errors, i.e. in the abstract an "error e" is used before "A total of 109....". Also the article would greatly profit from an english native speaker, who carefully checks the paper for better wordings and phrasings, i.g. "hypertensive urgencies" to name only 1 akwardness, needs to be "dismissed".
5. Why is "Furthermore" used in the abstract in line 30, if this does not relate to the sentence before?
6. Sentence in line 33: "with respect to KRd" is rather meant "as compared to"?
7. My conclusion - rather than a "closer and careful follow-up" - would be: a) Kd doses for more pretreated, often more elderly and comorbid (frail!) patients with Kd rather than KRd need to be considered very carefully, b) always and repeatedly checked, if not given in too high in doses and c) should be started slowly with rather increasing doses than with high doses at treatment start, d) Kd doses of 56mg/m2 for these need to be tappered down for best cardiac tolerance and expecially with adequate MM response, if cardiac events do occur, e) Kd and Krd patients - as asssessed here - were much different, therefore their direct comparison - albeit of interest to perform - biased due to substantial pt differences and should therefore rather b avoided than encouraged, f) fraily follow-up assessments in these patents, according to Holler,.....Engelhardt, Haematologcia 2023, Mian, Blood Cancer Journal 2023 or Larocca are especially of interest in these Cfz-patients to see whether they improve in pt performance of deteriorate and drug doses need to be adapted accordinly.
8. The authors should involve a very knowledgeable statistician and correct for main differences in the Kd and KRd group, namely for a. more advanced age in the Kd group, b. more pretreatment, c. longer MM duration, d. more advanced ISS 3 stage, e. CV diseases that were in almost all domains increased in the Kd group and f. LVEF that was decreased in the Kd group likewise and - most essentially: g. K-doses - and then recheck and reevaluate, whether differences in both groups may not be similar.
9. In Fig. 1, the authors should include percentages of subgroup assessment, i.e. the 2. assessment in n=88 pts was performed in 81%, whereas hem reports and phone calls in n=109 (100%).
10. In Table 1, median and mean (plus ranges) of Cfz-doses must be displayed, which should have been approx. 56 in the Kd and 27 in the KRd group.
If there are missings, i.e. in the ISS stage groups, than the authors cannot just stae, that these were 25 and 20 in Kd and KRd groups and calculate the percentages of those few, where ISS I/II vs. III were available, but give percentages according to pt numbers of ISS I/II, III and missings.
11. In Fig. 2, the x- and y-axis and Fig. descriptions in general need to be increased in font size from approx. 6-7 to readable sizes for much better readability.
12. Table "2" on page 8 is really "Table 3".
13. In Fig. 3, the authors should add an arrow according to "K. Jordan" and Engelhardt" papers that would indicate in each of the 4 subgraphs in Fig. 3, if the box blot increases of decreases in the Krd group indicate "for the better" or "for the worse".
14. A table accordig to beautiful S.Bringhen papers of general Cfz-recommmendation, namely what to do the Kd or Krd treated patients, if cardiac events do occur, whould be great to add as a Fig. 3, namely: to assess i.e.
a) frailty, i.e. whether that has improved or did deteriorate over time (see recent paper by Holler, .....Engelhardt et al. in Haematologica 2023),
b) if frailty increases over time (according to Holler et al. in only 10% of MM pts!) and/or cardiac events occur: reduce the Cfz dose,
c) monitor pts closely, if b) occurs,
d) if MM response is rewarding (>PR): and cardiac events do persist: reduce Cfz further or even pause etc. to make sure, that SAEs are avoided,
-> that Fig. would therefore ensure that MM experts do profit from the paper as much as possible.
Author Response
We thank the reviewer for the suggestions, please see the attachment.

Reviewer 2 Report
The Authors performed this study with the aim of investigating the difference in incidence and time of onset of cardiovascular adverse events between Kd and KRd in MM patients in real-life conditions. The paper reported that Kd regimen showed a greater and earlier cardiovascular toxicity with respect to KRd, but the relatively small sample of the cohort without a control arm and the absence of standardized dosing of the k containing regimen, as correctly reported by the Authors, does not allow to obtain robust results. However, this paper supports the central role of a careful cardiovascular assessment and close monitoring in patients in whom are planned regimens containing high doses of k and this is an useful information for clinicians who treat MM patients.
Recommandations for revision:
Abstract: It is clear and well written
Introduction: The Authors should better expalin the rationale for comparing two regimens containing the same combination of drugs (Kd), possibly the different dosage of k.
Materials and Methods: The Methods used are appropriate.
Results: The relatively small number of patients of a single Institution prevents you from getting robust results and achieving in several subgroups of analysis the statistical significance. Please, when you show the results report this point.
Table and Figures: They are clear and well detailed.
Discussion: The Authors should extend the section regarding the limitations of the study. In particular they should discuss more in detail and critically the impact of these limitations on the results reported in the study.
Author Response

(The authors gave the same response as above.)

Round 2
Reviewer 2 Report
The Authors have modified the text by including the suggestions of the referee. In the present form the paper is clearer and more acceptable.
Author Response
We thank the reviewer for the revision and the suggestions that have improved our work